# Nonlinear Behavior of High-Intensity Ultrasound Propagation in an Ideal Fluid

**Jitendra A. Kewalramani [1],\***, **Zhenting Zou [2]**, **Richard W. Marsh [3]**, **Bruce G. Bukiet [4]** **and Jay N. Meegoda [1]**

[1] Department of Civil & Environmental Engineering, New Jersey Institute of Technology, Newark, NJ 07102, USA; meegoda@njit.edu

[2] Dynamic Engineering Consultants, Chester, NJ 07102, USA; zz85@njit.edu

[3] Department of Chemical & Materials Engineering, New Jersey Institute of Technology, Newark, NJ 07102, USA; rwm26@njit.edu

[4] Department of Mathematical Science, New Jersey Institute of Technology, Newark, NJ 07102, USA; bukiet@njit.edu

\* Correspondence: jak93@njit.edu

**Abstract:** In this paper, nonlinearity associated with intense ultrasound is studied by using the one-dimensional motion of nonlinear shock wave in an ideal fluid. In nonlinear acoustics, the wave speed of different segments of a waveform is different, which causes distortion in the waveform and can result in the formation of a shock (discontinuity). Acoustic pressure of high-intensity waves causes particles in the ideal fluid to vibrate forward and backward, and this disturbance is of relatively large magnitude due to high-intensities, which leads to nonlinearity in the waveform. In this research, this vibration of fluid due to the intense ultrasonic wave is modeled as a fluid pushed by one complete cycle of piston. In a piston cycle, as it moves forward, it causes fluid particles to compress, which may lead to the formation of a shock (discontinuity). Then as the piston retracts, a forward-moving rarefaction, a smooth fan zone of continuously changing pressure, density, and velocity is generated. When the piston stops at the end of the cycle, another shock is sent forward into the medium. The variation in wave speed over the entire waveform is calculated by solving a Riemann problem. This study examined the interaction of shocks with a rarefaction. The flow field resulting from these interactions shows that the shock waves are attenuated to a Mach wave, and the pressure distribution within the flow field shows the initial wave is dissipated. The developed theory is applied to waves generated by 20 KHz, 500 KHz, and 2 MHz transducers with 50, 150, 500, and 1500 W power levels to explore the effect of frequency and power on the generation and decay of shock waves. This work enhances the understanding of the interactions of high-intensity ultrasonic waves with fluids.

**Keywords:** power ultrasound; nonlinear wave propagation; shock; rarefaction

---

## 1. Introduction

Ultrasound is a type of sound wave with frequencies ranging from 20 KHz up to several gigahertz. For practical applications, ultrasound is mostly generated by ultrasonic transducers and propagates into the subject material. Ultrasonics is a branch of acoustics dealing with the generation and use of inaudible sound waves [1]. Applications of ultrasonics are rigidly classified as being of either low intensity (popularly known as non-destructive applications) or high-intensity (also known as power ultrasonics) [1]. A low-intensity acoustic wave in a homogeneous medium propagates at a certain speed (relative to the medium), and deformations produced by the wave in the medium are purely elastic. Ultrasonic non-destructive testing and imaging used as means of exploration, detection,

and information (e.g., the location of a crack or determination of material properties), are some of the promising low-intensity applications.

High-intensity acoustic wave behaves differently than low-intensity: they can permanently change the physical, chemical, or biological properties or, if intense enough, even destroy the medium to which it is applied [1]. At high intensity, the amplitudes of vibration used are sufficiently high that nonlinear propagation occurs. The uses of high-power ultrasonics include cleaning, enhancing chemical reactions, emulsification, dispersion, welding of metals and polymers, machining and metal forming in solids and fluids, food processing, ultrasonic agglomeration, etc. The nonlinear nature of intense acoustic waves underlies many applications of high-intensity acoustic waves [2,3]. The branch of acoustics dealing with high-intensity acoustic is called nonlinear acoustics.

In ordinary elastic media, finite-amplitude sound propagation has been intensively studied to understand many nonlinear acoustic effects such as cumulative wave distortion with propagation distance as well as the concept of the parametric array [4]. Nonlinear acoustics is based on a nonlinear theory of elasticity [5]. Once the original sinusoidal finite-amplitude wave propagates from an ultrasonic source into a fluid medium, the wave will be exposed to two effects influencing its time course: (1) the dissipation arising from viscosity, heat conductivity, and relaxation processes and (2) the nonlinearity leading to the formation of higher harmonics to the fundamental frequency of the wave [6]. Understanding the basic mechanisms of the nonlinear effects of intense acoustic waves is critical to studying the distribution of high-intensity acoustic waves in a system [7]. The purpose of this paper is to understand and model the nonlinear behavior of high-intensity acoustic waves. The nonlinearity associated with intense ultrasound is studied by using the one-dimensional motion of nonlinear shock wave in an ideal fluid.

## 2. Nonlinear Wave Propagation and Shock Formation

In linear wave motion, disturbances in a medium can be defined as the result of three independent modes: the acoustic, entropy, and vorticity modes [2,8]. If wave intensity increases, the disturbances in a medium becomes large enough such that nonlinear wave propagation starts [3,7]. The nonlinear elastic behavior of the materials becomes progressively more important at high-amplitude wave excitation [3,9]. As the speed of the disturbance (or particle velocity) ($u$) increases, linear wave motion ceases, and therefore the superposition principle is no longer valid. Thus, the disturbance cannot be described in the form of the three independent modes [8]. These modes start interacting with one another. As shown in Figure 1, the following three interactions between these modes are activated due to high-intensities: the sound-sound, sound-vorticity and sound-entropy interactions [2,8]. Due to the sound-sound interactions, harmonic generations and self-demodulations occur within the acoustic mode [2,8]. Then, the acoustic mode starts to interact with the other two modes, and this interaction induces acoustic heating and generation of hydrodynamic flow [8].

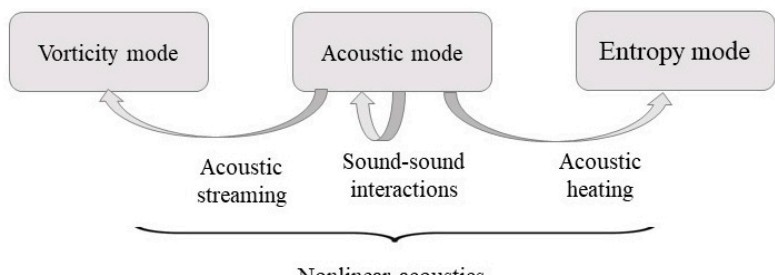

**Figure 1.** Study of nonlinear acoustics [Adopted based on a figure in Sapozhnikov (2015)].

As a consequence of nonlinearity, the local sound speed $c$ depends on the particle velocity $u$ ($c$ is function of $u$), $c_u$, and also the local wave propagation velocity depends on the particle velocity $u$ [2]. Thus, there are two independent sources of the acoustic nonlinearity:

- Change of the wave propagation speed due to drift with velocity $u$.
- Change in local sound speed from $c_0$ to $c_u$.

This transition from the linear regime to a nonlinear corresponds to a change in the local wave propagation from $c_0$ to $c_u + u$, as shown in Figure 2. The sections of the waveform with higher $u$ propagate faster than those with lower $u$. This accumulation of the nonlinear effects in the waveform leads to variation (or deviation) in the propagation speed of different segments of the waveform (Figure 2c) resulting in distortion of the sinusoidal wave and increasing steepness of the wavefront (Figure 2d). Eventually, these distortions result in a shock formation represented by an N-wave (Figure 2e) [3,7,10]. At the shock, the jump condition forms, and the medium undergoes an abrupt and nearly discontinuous change in the pressure, density, and temperature [2,4,10]. The shock formation corresponds to the first appearance of infinitely steep waveform regions associated with the N-wave. The critical distance $\dot{X}$ is the distance propagated by an acoustic wave in an ideal medium when the infinite slope first appears in the waveform [7].

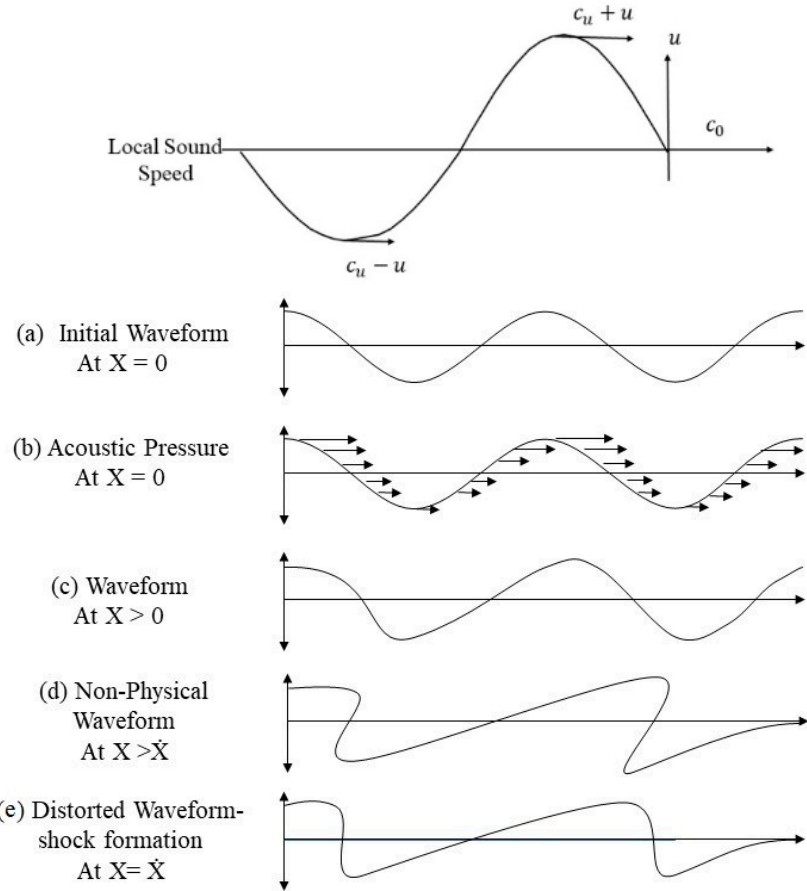

**Figure 2.** Schematic representation of acoustic distortion and shock formation during nonlinear propagation [Adopted based on figures in Blackstock and Hamilton (1998) and Leighton (2007)].

## 3. Nonlinear Interactions Within the Acoustic Mode

In order to analytically study the nonlinear behavior of intense ultrasound waves, an ideal model is to consider simulating a sinusoidal wave propagating in one-dimension. The motion of the wave can be modeled as the flow of an ideal fluid in an infinitely long tube extending along the *x*-axis, with one end having a moving piston and the other end being either open or having a fixed wall, where there is no viscosity or thermal conductivity [10]. In such a case, $v = (u, 0, 0)$ and all variables depend only on *x* and *t*.

In a gas-filled tube, if a piston is moved into it or if a receding piston is stopped a shock (discontinuities) is generated that moves away from the piston [10]. Similarly, a forward-moving

rarefaction wave is sent into the medium when the piston recedes away from the fluid [10]. Rarefactions and shocks have finite speed such that the regions they influence grow over time. A pulse high-intensity ultrasonic wave can be conceived as one complete cycle of piston motion in a tube. That is, the piston moves from a stationary or a rest position at a constant speed until it reaches the maximum point of displacement, stops, and reverses direction, moving in the opposite direction at the same speed as earlier until it reaches its original position and stops again. Figure 3 shows the generation of shock and rarefaction waves, where (a) a shock is produced by a piston moving with constant velocity into the fluid at rest, (b) the piston reaches its maximum displacement, changes its direction of propagation, and the forward-moving rarefaction wave is sent into the compressed fluid behind the shock, and (c) when the piston comes to rest, another forward moving shock is produced.

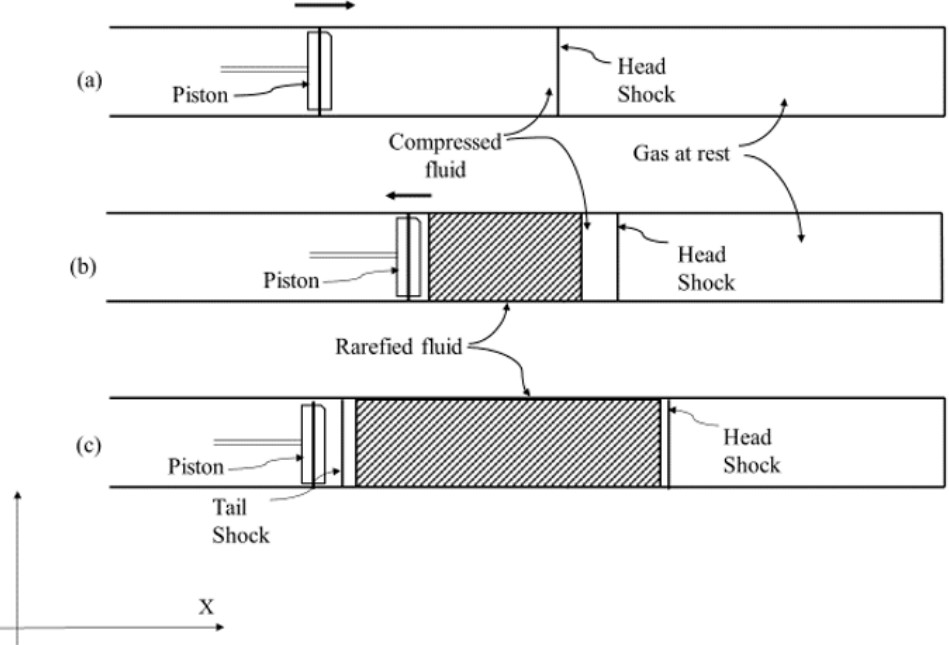

**Figure 3.** Generation of shock and rarefaction waves [Adopted based on a figure in Courant, (1948)].

The one-dimensional flow eliminates the vorticity mode, and the absence of dissipations makes the process adiabatic. That is, one may exclude the entropy modes as well. Therefore, a 1-D model describes only the nonlinear behavior of the acoustic mode [4]. The propagation of intense acoustic waves in one-dimensional motion can be mathematically described by using the conservation equations of mass, momentum, and energy [11]:

$$\begin{pmatrix} \rho \\ m \\ e \end{pmatrix}_t + \nabla. \begin{pmatrix} m \\ \frac{m^2}{\rho} + P \\ \left(\frac{m}{\rho}\right)(e + P) \end{pmatrix} = 0 \tag{1}$$

where $\rho$ is the density of the fluid, $m$ is the momentum, $P$ is the pressure, $\nabla.$ is the gradient, and $e$ is the energy per unit volume which is expressed as:

$$e = \rho\varepsilon + \frac{\rho u^2}{2} \tag{2}$$

where $u$ is the particle velocity, and specific internal energy $\varepsilon$ is given by:

$$\varepsilon = \frac{P}{\rho(\gamma - 1)} \tag{3}$$

where $\gamma$ is a specific heat ratio and with shock speed $U$, the conservation of mass can be written as:

$$\rho_1(u_1 - U) = \rho_0(u_0 - U) \tag{4}$$

The subscripts 0 and 1 represent the constant states on each side of the shock. The conservation of momentum for the shock can be expressed as:

$$\rho_1(u_1 - U)^2 + P_1 = \rho_0(u_0 - U)^2 + P_0 \tag{5}$$

The conservation of energy for the shock can be written as:

$$(u_1 - U)(e_1 + P_1) = (u_0 - U)(e_0 + P_0) \tag{6}$$

By using the specific volume, $\tau = 1/\rho$, Equations (4)–(6) can be reduced to the Hugoniot relation [12]:

$$\frac{\gamma_0 \tau_0 P_0}{\gamma_0 - 1} - \frac{\gamma_1 \tau_1 P_1}{\gamma_1 - 1} = \frac{(P_0 - P_1)(\tau_0 + \tau_1)}{2} \tag{7}$$

Equations (4)–(6) describe the nonlinear behavior of the acoustic mode alone. Riemann showed these equations can have an exact general solution in the form of shock, rarefaction, and contact wave across which density and pressure changes [10]. The solution would consist of a right propagating wave, a left propagating wave, and a contact wave [12]. The right (or left) wave can be a shock or a rarefaction. The intermediate region can be connected to left or right regions by following Riemann invariants and using the isentropic law [12]:

$$\begin{array}{c} \frac{u_L}{2} + \frac{c_L}{\gamma_L - 1} = \frac{u_*}{2} + \frac{c_*}{\gamma_* - 1} \\ \frac{u_R}{2} + \frac{c_R}{\gamma_R - 1} = \frac{u_*}{2} + \frac{c_*}{\gamma_* - 1} \end{array} \tag{8}$$

$$\begin{array}{c} P_L \rho_L^{-\gamma_L} = P_* \rho_*^{-\gamma_*} \\ P_R \rho_R^{-\gamma_R} = P_* \rho_*^{-\gamma_*} \end{array} \tag{9}$$

where * denotes the region inside the wave or mid-region. The equations for shock or rarefaction waves can be found by eliminating $U$ and $\rho_1$ from Equations (4)–(6). Since the left and right regions connect to regions with the same velocity and pressure, the equations can be expressed in terms of velocity as a function of pressure [12]. Newton's iteration scheme is used to solve the equation for the velocity and pressure of the mid-region. Bukiet, 1988, presented the following equations, relating the intermediate state of the wave to the side state [12]:

Right shock:

$$u_* = u_R + \frac{A_R(\alpha_R - 1)}{(D_R \alpha_R + E_R)^{1/2}}, \frac{du}{dP} = \frac{\alpha_R D_R + 3\gamma_R - 1}{\sqrt{2\rho_R P_R}(D_R \alpha_R + E_R)^{3/2}} \tag{10}$$

Right rarefaction:

$$u_* = u_R + \frac{2C_R(\alpha_R{}^{B_R} - 1)}{E_R}, \frac{du}{dP} = \frac{1}{H_R \alpha_R{}^{J_R}} \tag{11}$$

With:

$$\alpha_R = \frac{P}{P_R}; A_R = \sqrt{\frac{2P_R}{\rho_R}}; B_R = \frac{\gamma_R - 1}{2\gamma_R}; C_R = \sqrt{\frac{\gamma_R P_R}{\rho_R}}; D_R = \gamma_R + 1; E_R = \gamma_R - 1;$$
$$H_R = \sqrt{\gamma_R P_R \rho_R} J_R = (\gamma_R + 1)/2\gamma_R$$

where $R$ denotes the region of fluid to the right or left side of mid-region.

Equations (4) and (7) can be used to compute mid-region density and wave speed for the shock, while Equations (8) and (9) can be used for rarefactions.

### 3.1. Head Shock

With the one-dimensional model, as the intense ultrasound wave is sent into a medium, using the piston model analogy, a shock will be generated. As mentioned earlier, a nonlinear wave propagates in one direction only, assuming the direction of propagation is to the right, the right shock Equation (15) can be used to calculate the pressure jump across the shock. To demonstrate the calculation, the following parameters for an ultrasound transducer were assumed: a transducer of 1500 W with 80% efficiency, source diameter of 12.7 mm, frequency of 2 MHz, and an ideal fluid where $\rho = 1.225 \text{kg/m}^3$ and $c_0 = 343$ m/s. For the given parameters, the amplitude of displacement, $A$, will be $1.69 \times 10^{-5}$ m and the displacement speed $u$ will be 212.39 m/s. (*Calculations for $\rho_{r1}$ and $u$ are shown in Appendix A).

As shown in Figure 3 the fluid particles in the front of the head shock are in a calm state; the particle speed is 0 m/s, and behind the shock, the particle speed (or piston) is 212.39 m/s. Equation (10) is used to calculate the pressure behind the jump. In the equation below the subscripts "1" and "r1" represent the fluid ahead of and behind the shock, respectively. Therefore, $u_1$ and $c_1$ are equal to 0 and 343 m/s, respectively, as the fluid in front of the head shock is undisturbed, $P_1$, the atmospheric fluid pressure, is equal to 101,325 Pa, $\rho_1$ is equal to the 1.225 kg/m$^3$, and the specific heat ratio $\gamma$ for the fluid is 1.4. The particle speed behind the head shock is $u_{r1}$ is 212.39 m/s. The unknown $P_{r1}$ can be calculated using Equation (10):

$$u_{r1} = u_1 + \frac{\sqrt{\frac{2P_1}{\rho_1}\left(\frac{P_{r1}}{P_1} - 1\right)}}{\left[\left\{(\gamma+1)\frac{P_{r1}}{P_1}\right\} + (\gamma-1)\right]^{1/2}}$$

$$212.39 = 0 + \frac{\sqrt{\frac{2*101325}{1.225}\left(\frac{P_{r1}}{101325} - 1\right)}}{\left[(1.4+1)\frac{P_{r1}}{101325} + (1.4-1)\right]^{1/2}}$$

$$P_{r1} = 229,029 \text{ Pa } or \text{ } 40,530 \text{ } P$$

For the propagation direction of the piston as right, the particle speed $u_{r1}$ and sound speed $c_{r1}$ are in the same direction and $u_{r1} + c_{r1}$ is greater than $u_1 + c_1 (= 343$ m/s). Therefore, $P_{r1} > P_1$, so $P_{r1}$ is equal to 229,029 Pa. By applying the Hugoniot relationship, the density behind the head shock $\rho_{r1}$ is equal to 2.16 kg/m$^3$. Using the piston speed of particles as 212.39 m/s and the law of conservation, the head shock speed $U_1$ is 490.65 m/s[**]. Shocks always move at supersonic speed as observed from ahead of the shock, and subsonic speed as observed from behind the shock [10]. (** Calculations for $\rho_{r1}$ and $U_1$ are shown in Appendix B).

### 3.2. Rarefaction

As an acoustic wave propagates, fluid particles adjacent to the transducer (or piston) are vibrating back and forth in the direction of propagation around its original position. The reiteration of fluid particles can be modeled as the piston moving backward. In the second half of the cycle, the piston speed is the same as that it had in the forward (first half) cycle, but it becomes negative as the direction reverses and a forward propagating rarefaction wave is generated. A rarefaction is a smooth fan zone with continuously changing pressure, density, and velocity. In the equation below, the subscripts $r1$ and $r2$ represent the face and tail of rarefaction. The particle velocity at the rarefaction face is $u_{r1} = 212.39$ m/s. As the direction of piston reverses the velocity of the piston becomes negative, $u_{r2} = -212.39$ m/s. Using Equation (11) for the right rarefaction, $u_{r1} = 212.39$ m/s, representing

the speed attained by a particle at the face of piston with $P_{r1} = 229,029$ Pa and $\rho_{r1} = 2.16$ kg/m$^3$. $P_{r2}$, the pressure at the tail of rarefaction can be calculated as:

$$u_{r2} = u_{r1} + \frac{2\sqrt{\frac{\gamma * P_{r1}}{\rho_{r1}}}\left(\left(\frac{P_{r2}}{P_{r1}}\right)^{\frac{\gamma-1}{2\gamma}} - 1\right)}{\gamma - 1}$$

$$-212.39 = 212.39 + \frac{2\sqrt{\frac{1.4*229029}{2.16}}\left(\left(\frac{P_{r2}}{229029}\right)^{\frac{1.4-1}{2*1.4}} - 1\right)}{1.4 - 1}$$

$$P_{r2} = 40,055 \text{ Pa}$$

Equation (9) is used to determine density at the tail of the rarefaction wave,

$$P_{r2}\rho_{r2}{}^{-\gamma_R} = P_{r1}\rho_{r1}{}^{-\gamma_*}$$

$$40055 * \rho_{r2}{}^{-1.4} = 229029 \times 2.16^{-1.4}$$

$$\rho_{r2} = 0.62 \text{ kg/m}^3$$

For the rarefaction wave, the particle speed $u_{r2}$ and sound speed $c_{r2}$ are in opposite directions, and thus, $P_{r2} < P_1$. As mentioned before, the transition from the linear to the nonlinear regime corresponds to a change in propagation velocity from $c_0$ to $u + c_u$. Using this and the data from the shock speed calculations, the wave velocity at the rarefaction face is equal to $u_{r1} + c_{r1} = 212.39 + \sqrt{\frac{1.4*229029}{2.16}} = 212.39 + 385.69 = 597.68$ m/s, which is faster than the speed of the head shock. Similarly, the wave speed at the tail of the rarefaction wave will be $-u_{r2} + c_{r2} = -212.39 + \sqrt{\frac{1.4*40055}{0.62}} = -212.39 + 300.74 = 88.34$ m/s. With the knowledge of the state at the front and tail of the rarefaction, the intermediate states through the rarefaction can be computed [13].

To compute the pressure throughout the rarefaction, following constants $k1$, $k2$, and $k3$ can be assumed:

$$k1 = \frac{dx}{dt} = u_* + c_* \tag{12}$$

$$k2 = \frac{u_{r1}}{2} - \frac{c_{r1}}{\gamma - 1} = \frac{u_*}{2} - \frac{c_*}{\gamma - 1} \tag{13}$$

$$k3 = P_{r1}\rho_{r1}{}^{-\gamma} = P_*\rho_*{}^{-\gamma} \tag{14}$$

The values of k1, k2, and k3 can be related as follows, let k1 minus two k2 gives:

$$k1 - 2 * k2 = 6\sqrt{1.4 \times \frac{P_*}{\rho_*}} \tag{15}$$

From Equation (14),

$$P_* = k3 * \rho_*^{1.4} \tag{16}$$

Substituting P* from Equation (16) in Equation (15) to obtain:

$$\rho_* = \left[\frac{\left(\frac{k1-2k2}{6}\right)^2}{1.4k3}\right]^{2.5} \tag{17}$$

Here k2 and k3 are constant throughout the rarefaction. Therefore, knowing the state at the front rarefaction, and by using different values of k1, the density throughout the rarefaction can be found. Then, using Equations (17) and (9), the pressure throughout the rarefaction can be calculated as:

$$P_* = \frac{P_{r1} \times \rho_*^{1.4}}{\rho_{r1}^{1.4}} \tag{18}$$

Due to the smooth change in velocity through the rarefaction, $k1$ has its own velocity range between the tail rarefaction velocity of 88.3m/s and the face rarefaction velocity 597.7m/s. Consequently, $k2$ and $k3$ can be calculated as:

$$k2 = \frac{u_{r1}}{2} - \frac{C_{r1}}{\gamma - 1} = -857 k3 = P_{r1} \rho_{r1}^{-\gamma} = 77944$$

The rarefaction wave can be divided into two parts separated by a sound wave where $u_* + c_* = c_0$ [10]. Let $\rho_r$ and $P_r$ represent the density and pressure at the separation boundary, respectively. Using $k1 = u_r + c_r = 343 \ m/s$, and Equations (17) and (18) to find the physical state at separation boundary produces:

$$\rho_r = \left[ \frac{\left( \frac{343 + 2 * 857}{6} \right)^2}{1.4 * 77944} \right]^{2.5} = 1.21 \ \text{kg/m}^3$$

$$P_r = \frac{229029 \times 1.2^{1.4}}{2.16^{1.4}} = 101,161.4 \ \text{Pa}$$

Thus, the rarefaction can be divided into two zones, where $P_r = 101,161.4 \ \text{Pa}$ and $\rho_r = 1.21 \ \text{kg/m}^3$ at the boundary.

### 3.3. Tail Shock

At the end of the pulse wave, as the piston (and similarly) fluid particles come to rest, the displacement speed will change from $-212.39$ to $0 \ \text{m/s}$, and other forward-moving shocks will be generated. Performing calculations similar to the head shock pressure $P_2$, the density behind tail shock will be 99,954 Pa. Similarly, by applying the Hugoniot relation, the density $\rho_2$ behind the tail, shock equals 1.38 kg/m³ and the tail shock speed $U_2$ is 173.27 m/s [***]. ([***] Calculations for $P_2$, $\rho_2$, and $U_2$ are shown in Appendix C).

## 4. Decaying of N-Wave

As shown in Figure 4 a pulse of an ultrasonic wave consists of two shocks and rarefaction between them. The rarefaction can be divided into two parts separated by the sound wave, where $u_* + c_* = c_0$. The velocity of the head shock is subsonic relative to the local wave speed behind it, i.e., at the face of the rarefaction, the wave speed $u_{r1} + c_{r1}$ is faster than the head shock speed, $U_1$. When the rarefaction meets the shock, the shock is weakened, causing the shock speed to decrease. Similarly, the tail shock velocity is supersonic relative to the local sound speed ahead of it, and the tail shock will thus overtake the rarefaction wave. The forward part of the rarefaction weakens the head shock, and the tail shock reduces by the backward part of the rarefaction [10].

The speed of the waveform at the face of the rarefaction is $u_{r1} + c_{r1} = 597.7$ m/s, whereas the speed of head shock is $U_1 = 490.8$ m/s. Thus, after some time, the rarefaction will overtake the head shock causing the head shock to weaken. The rarefaction fan impinges on the head shock decaying it further. Similarly, the speed of the waveform at the tail of the rarefaction is $u_{r2} + c_{r2} = 88.3$ m/s, while the speed of the tail shock is $U_2 = 172.4$ m/s. Thus, the tail shock will overtake the backward face of the rarefaction wave, as shown in Figure 4.

In the 1950s, extensive experimental investigations were performed at the Institute of Aero physics, the University of Toronto (which was earlier known as Institute of Aerospace Studies) on the flow field resulting from one-dimensional wave interactions. Two studies namely 'On the One-Dimensional Overtaking of a Shock wave by Rarefaction waves' by Glass, et al., 1959 and another study 'On the One-Dimensional Overtaking of Rarefaction waves by a Shock wave' by Bermner et al., 1960 are used here to study the interaction between the head shock and the forward part of rarefaction wave and the tail shock and the backward region of rarefaction, respectively [14–16].

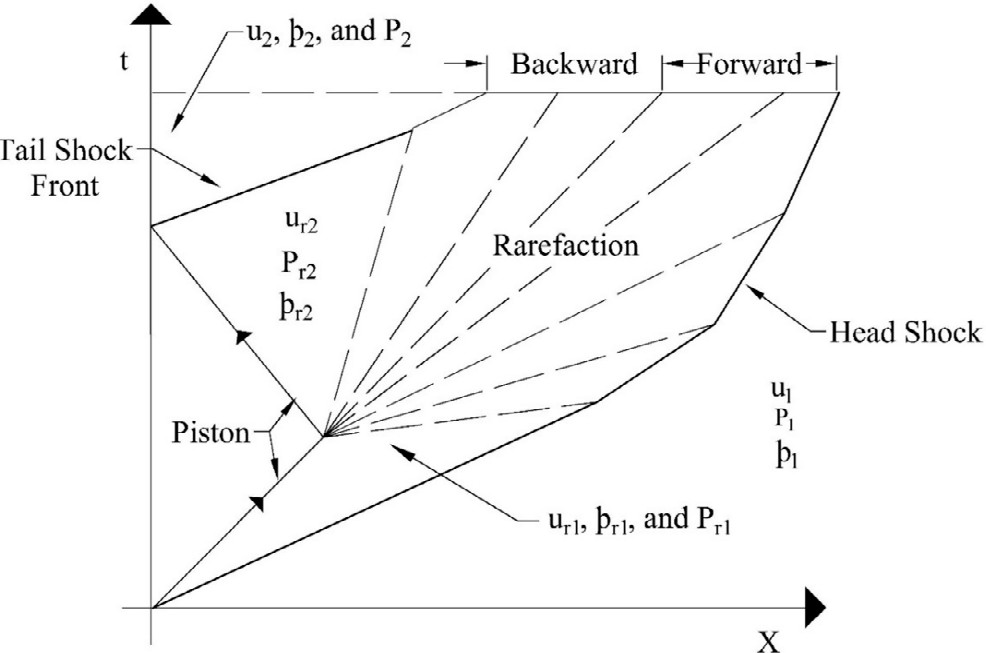

**Figure 4.** Decaying of N-wave [Adopted based on a figure in Courant, (1948)].

*4.1. Shock Wave Overtaken by a Rarefaction Wave*

The problem of the overtaking of a shock by a rarefaction wave has been considered by Courant and Friedrichs (1948) [10]. The possible wave systems that can result from such an interaction are as follows:

I.      A transmitted shock and a reflected rarefaction wave

II.     A transmitted shock and a reflected compression wave that steepens into a shock wave

III.    A transmitted rarefaction wave and a reflected rarefaction wave

IV.     A transmitted rarefaction wave and reflected compression wave that steepens into a shock wave

It is also possible to have the limiting cases where either or both the reflected or transmitted wave are Mach Waves.

In all these situations, the transmitted wave decays with interactions, its strength decreases, and the entropy change across the transmitted wave diminishes. Thus, there will be a region of entropy change, i.e., a contact region between the reflected and transmitted waves [14,15]. A Mach wave is the envelope of the wavefront, traveling at the sound speed propagated from an infinitesimal disturbance in supersonic speed. Glass et al., 1959 presented the algebraic expressions that give the final wave strength and states in terms of the initial wave strengths and specific heat ratio [15].

The strength of a shock (or rarefaction) wave is defined as the ratio of fluid pressure ahead of and behind the wave. The final waves and its states resulting from the interaction in a one-dimensional flow of a perfect gas are dependent on the relative strengths of the initial interacting shock $(P_1/P_5)$ and rarefaction waves $(P_5/P_4)$ [14,15]. For interaction, the overtaking wave $(P_5/P_4)$ can be defined as weak or strong, based on the ratio of the pressure jump (or relative strength of) across the overtaken wave $(P_1/P_5)$ and overtaking waves $(P_5/P_4)$, i.e., $(P_1/P_5)/(P_5/P_4) = P_1/P_4$. This ratio can also be denoted as $P_{14}$ [15]. The overtaking wave in this system of interactions is considered to be a weak shock if the ratio is less than or equal to 1, $(P_{14} \leq 1)$, and is considered as strong if the ratio is more than 1, $(P_{14} > 1)$.

Figure 5 shows (A), the overtaking of a shock by a weak rarefaction wave and the flow field resulting from this interaction can result in (B) a transmitted shock wave with a reflected rarefaction or (C) a transmitted shock with a reflected wave that can be a shock, with a contact region between the reflected and transmitted waves. As this transmitted wave decays with interactions, its strength

decreases, and the entropy diminishes. Glass et al., 1959 presented algebraic equations relating the pressure ratio across the transmitted and reflected wave (at the end of interactions) to the known values of pressure ratios across the incident shock and overtaking rarefaction wave (before interactions) [15]. When the overtaking rarefaction wave is weak, the interaction results in a reflected rarefaction wave, the quasi-steady flow in regions $(P_2)$ and $(P_3)$ can be evaluated using expressions Equations (19) and (20) [14]. Here, the reflected compression wave (case II, Figure 5C) was tactically assumed. Hence, it would not influence the terminal states and can be neglected [15]. Case III and case IV mentioned above correspond to the strong overtaking rarefaction wave and they do not pertain to our discussion. Thus, in the case of a weak rarefaction wave overtaking a shock wave, the resulting flow field or terminal state will consist of a transmitted shock and a reflected rarefaction wave (case I, Figure 5B). The pressure jump across it can be evaluated using Equations (19) and (20).

$$\sqrt{\frac{\beta P_{15}(\alpha + P_{15})}{(1 + \alpha P_{15})}}\left[\frac{1 - P_{15}}{\sqrt{P_{15}(\alpha + P_{15})}} + \frac{1 - P_{21}}{\sqrt{\alpha P_{21} + 1}}\right] + 2(P_{45})^{\beta} - (P_{15}P_{21})^{\beta} - 1 = 0 \qquad (19)$$

$$(P_{34}) = P_{21}P_{15}P_{54} \qquad (20)$$

where, $\beta = \frac{\gamma - 1}{2\gamma}$ and $\alpha = \frac{\gamma + 1}{\gamma - 1}$.

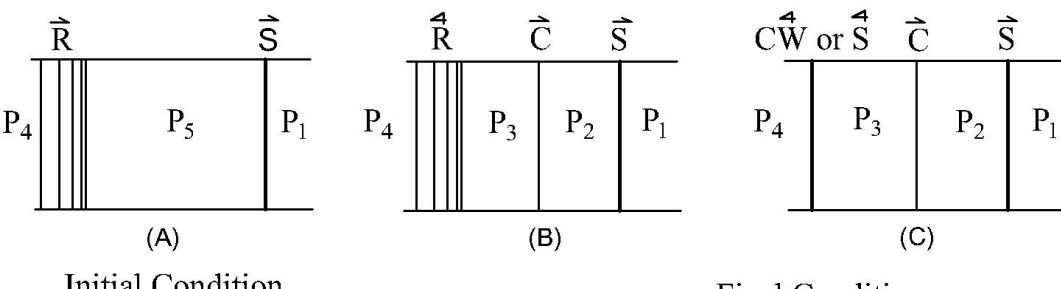

**Figure 5.** Overtaking of a shock wave by a weak rarefaction wave [Adopted based on a figure in Glass et al (1959)].

## 4.2. Shock Wave Overtaking a Rarefaction Wave

Bermner et al., 1960 presented an analytical solution for a 1-D shock wave overtaking a rarefaction wave. Analytical solutions are presented such that if the initial strength of the rarefaction wave and overtaking shock wave strength are known, it is possible to predict the strengths of the reflected and transmitted wave after the interaction as well as the properties of the newly-formed quasi-steady regions.

As in the previously described case, here, the resulting wave pattern depends on the relative strength of the initial interacting shock wave and the rarefaction wave. Figure 6 shows a schematic representation for a relatively weak shock overtaking a rarefaction wave, i.e., $(P_{14} \leq 1)$, where (A) a shock of known strength overtaking a rarefaction wave, (B) the flow field resulting from the interaction can result in a reflected shock wave with a reflected rarefaction or (C) the reflected and transmitted waves both being rarefaction waves. Bermner et al., 1960 presented algebraic equations relating the pressure ratio across the transmitted and reflected wave (at the end of interactions) to the known values

of pressure ratios across the incident rarefaction and overtaking shock wave (before interactions) [16]. For a weak incident shock wave, (a case when it completely attenuates to a Mach wave), the reflected wave is a shock wave and the quasi-steady flow in the regions $(P_2)$ and $(P_3)$ in Figure 6 can be evaluated using Equations (21) and (22) [16]. Here the reflected rarefaction wave (Figure 6C) was tactically assumed. Hence, it would not influence the terminal states and can be neglected [14]. Thus, in case of a weak shock wave overtaking a rarefaction wave, the resulting flow field or terminal state will consist of a transmitted rarefaction and a reflected shock wave. The pressure jump across them can be evaluated using Equations (21) and (22).

$$\frac{1-(P_{34})^{\beta}(P_{45})^{\beta}}{\sqrt{\beta}}+\frac{(P_{45}-1)}{[1+\alpha(P_{45})]^{1/2}}+\left[\frac{P_{45}(\alpha+P_{45})}{1+\alpha P_{45}}\right]^{1/2}\frac{(1-P_{34})}{[1+\alpha P_{34}]^{1/2}}=0 \quad (21)$$

and

$$(P_{34})=\frac{P_{21}}{P_{45}P_{51}} \quad (22)$$

where, $\beta=\frac{\gamma-1}{2\gamma}$ and $\alpha=\frac{\gamma+1}{\gamma-1}$.

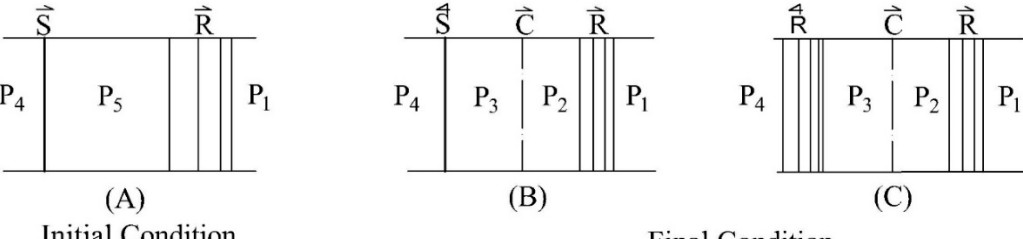

(A)       (B)       (C)

Initial Condition      Final Condition

$\vec{R}\ \vec{S}$   Forward facing rarefaction wave and shock wave

$\overleftarrow{R}\ \overleftarrow{S}$   Backward facing rarefaction wave and shock wave

$C\overleftarrow{W}$   Backward facing compression wave

$\vec{C}$       Contact regions

P       Pressure

**Figure 6.** Overtaking of a rarefaction wave by a weak shock wave [Adopted based on a figure in Glass et al (1959)].

Based on the above discussion, for a head shock overtaken by the forward part of rarefaction $\left(\frac{P_1}{P_r}\le 1\right)$, the resulting flow field can be evaluated using Equations (19) and (20):

$$\sqrt{\frac{\beta\frac{P_1}{P_{r1}}\left(\alpha+\frac{P_1}{P_{r1}}\right)}{\left(1+\alpha\frac{P_1}{P_{r1}}\right)}}\left[\frac{1-\frac{P_1}{P_{r1}}}{\sqrt{\frac{P_1}{P_{r1}}\left(\alpha+\frac{P_1}{P_{r1}}\right)}}+\frac{1-\frac{P_{r12}}{P_1}}{\sqrt{\alpha\frac{P_{r12}}{P_1}+1}}\right]+2\left(\frac{P_r}{P_{r1}}\right)^{\beta}-\left(\frac{P_{r12}}{P_1}\frac{P_1}{P_{r1}}\right)^{\beta}-1=0$$

and

$$\left(\frac{P_{r13}}{P_r}\right)=\frac{P_{r12}}{P_1}\frac{P_1}{P_{r1}}\frac{P_{r1}}{P_r}$$

Using the above expression, $P_{r12}=101,269$ Pa and $P_{r13}=101,269$ Pa.

Similarly, using Equations (21) and (22), evaluating the flow field resulting from the interaction of the tail shock wave overtaking the backward part of the rarefaction wave can be shown by:

$$\frac{1 - \left(\frac{P_{r23}}{P_2}\right)^{\beta}\left(\frac{P_2}{P_{r2}}\right)^{\beta}}{\sqrt{\beta}} + \frac{\left(\frac{P_2}{P_{r2}} - 1\right)}{\left[1 + \alpha\left(\frac{P_2}{P_{r2}}\right)\right]^{1/2}} + \left[\frac{\frac{P_2}{P_{r2}}\left(\alpha + \frac{P_2}{P_{r2}}\right)}{1 + \alpha\frac{P_2}{P_{r2}}}\right]^{1/2} \frac{\left(1 - \frac{P_{r23}}{P_2}\right)}{\left[1 + \alpha\frac{P_{r23}}{P_2}\right]^{1/2}}$$

and

$$\left(\frac{P_{r23}}{P_2}\right) = \frac{P_{r22}}{P_r}\frac{P_r}{P_{r2}}\frac{P_{r2}}{P_2}$$

Using the above expression, $P_{r12} = 100,527$ Pa *and* $P_{r13} = 100,527$ Pa.

Figure 7 shows the resulting pressure across states ($P_{r12}$–$P_{r23}$) and transmitted shocks ($P_1$ and $P_2$). After the interaction, the state behind the head shock and ahead of forwarding rarefaction have the same pressure. Thus, the forward rarefaction has attenuated the head shock to a Mach wave, just as its tail reaches the head shock. Similarly, in the case of the tail shock overtaking the backward rarefaction, states ($P_{r22}$–$P_{r23}$) and ($P_2$) are similar pressure. Thus, the incident tail shock has been attenuated to a Mach wave, just as it reaches the head of the backward rarefaction [16]. Thus, after the interaction pressure jump across head shock becomes nearly 1 ($101,269/101,325 = 0.99 \sim 1$) which before the interaction was 2.26 ($229,029/101.325$), similarly pressure jump across tail shock becomes unity ($99,954/100,527 = 0.9943 \sim 1$) which before the interaction was 2.46 ($99,954/40,055$).

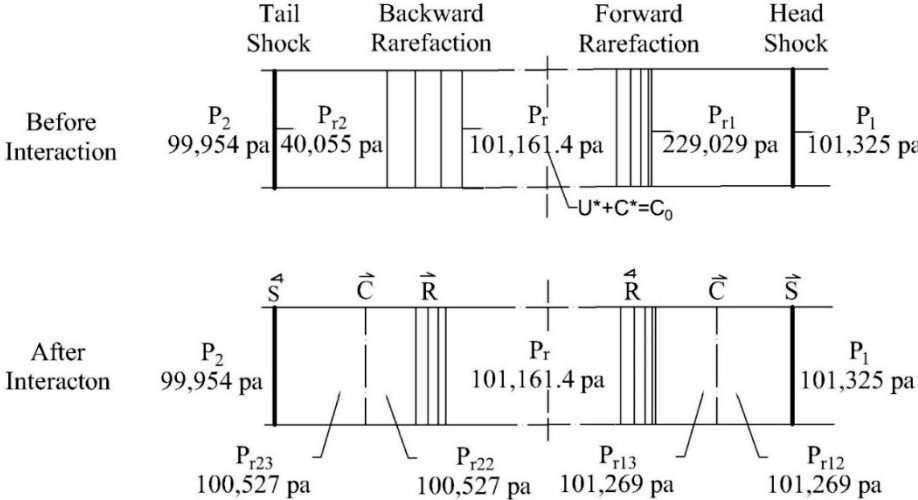

**Figure 7.** Flow field resulting from interactions.

Both the shocks continue to decay during these interactions with the rarefaction waves, and their strength (the pressure jumps across it) diminishes. Thus, at some distance from the source, both the shock at head and tail of the N-wave get attenuated to Mach waves. As entropy across the shock continues to decay due to interactions with the rarefaction, there is a reduction in the waveform speed and amplitude. This reduction is due to the dissipation of ultrasound energy. The conservation of energy implies this dissipated energy is transferred from one form to another. In this case, it becomes heat.

A similar set of calculations is performed for transducers of power levels of 50, 150, 500, and 1500 W and for frequencies of 20, 500 kHz and 2 MHz, as shown in Tables 1 and 2 and results are presented in Figure 8.

**Table 1.** Vibration speed of fluid particle in acoustic field of varying power level and frequency.

| Power Level, W (watt) | Frequency, f (kHz) | Intensity, I (W/m$^2$) | Amplitude, A ($10^{-6}$ m) | Particle Speed, V (m/s) |
|---|---|---|---|---|
| 1500 | 20 | 9,477,702.96 | 1691 | 212.39 |
| 1500 | 500 | 9,477,702.96 | 67.6 | 212.39 |
| 1500 | 2000 | 9,477,702.96 | 16.9 | 212.39 |
| 500 | 20 | 3,152,941.98 | 976.3 | 122.61 |
| 500 | 500 | 3,152,941.98 | 39.1 | 122.61 |
| 500 | 2000 | 3,152,941.98 | 9.76 | 122.61 |
| 150 | 20 | 94,772.6 | 534.7 | 67.22 |
| 150 | 500 | 94,772.6 | 21.4 | 67.2 |
| 150 | 2000 | 94,772.6 | 5.34 | 67.22 |
| 50 | 20 | 315,924.2 | 308.7 | 38.78 |
| 50 | 500 | 315,924.2 | 12.3 | 38.78 |
| 50 | 2000 | 315,924.2 | 3.08 | 38.78 |

**Table 2.** Variation in resulting flow field.

| Particle Speed, V(m/s) | $P_1$ (Pa) | $P_{r1}$ (Pa) | $P_r$ (Pa) | $P_{r2}$ (Pa) | $P_2$ (Pa) | $P_{r12} = P_{r13}$ (Pa) | $P_{r22} = P_{r23}$ (Pa) | Head Shock, $U_1$ (m/s) |
|---|---|---|---|---|---|---|---|---|
| 212.39 | 101,325 | 229,029 | 101,161 | 40,055 | 99,954 | 101,269 | 100,527 | 490.8 |
| 122.61 | 101,325 | 164,678 | 102,005 | 60,027 | 101,058 | 102,004 | 101,152 | 421.7 |
| 67.22 | 101,325 | 132,853 | 102,220 | 76,414 | 101,281 | 102,228 | 101,294 | 383 |
| 38.78 | 101,325 | 118,629 | 102,258 | 86,225 | 101,317 | 102,260 | 101,319 | 364 |

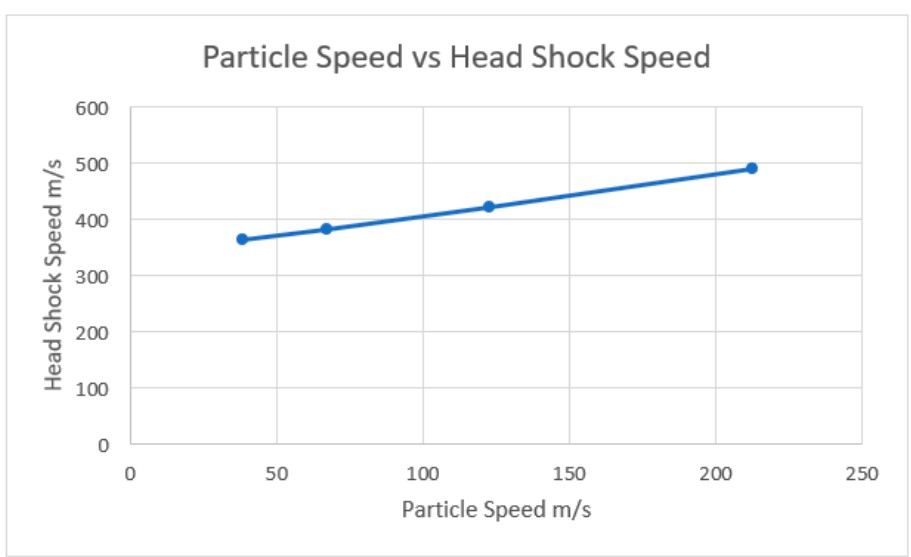

**Figure 8.** Particle speed vs. head shock.

## 5. Discussions of Results

In an ideal fluid, the displacement amplitude of an intense ultrasonic wave decreases with an increase in frequency, as amplitude is inversely proportional to the intensity of the disturbance or vibration. However, the vibration speed remains constant for a given power level for different frequencies as shown in Table 1. The speed of the head shock and thereby the strength of the shock depends on the disturbance speed of particle as shown in Figure 8, where the higher the vibration speed, the stronger the discontinuity. For example, for a particle speed of 212.39 m/s, the strength of the head shock is $P_{r1}/P_1$ which equals 2.26, whereas for a particle speed of 122.61 m/s, $P_{r1}/P_1$ is 1.62. Similarly, it can be seen from the pressure distribution in the resulting flow field (before the interaction) that the strength of the wave will also increase with an increase in the power level. After the interaction

of the rarefaction wave and the shock, the shock strength (the pressure jumps across it) diminishes. For instance, for particle speed of 212.39 m/s the pressure jump across the head shock was 2.26 before the interaction and reduces to near atmospheric after the interaction. This intersection causes dissipation of ultrasound energy into heat. Figure 9 summarizes this observation graphically. Figure 9 shows the initial and final conditions of the N-Wave, where (a) in a pulse of intense ultrasonic waves, two shocks will form with a rarefaction wave in between due to the nonlinear wave motion of the wave, (b) the rarefaction can be divided into two parts separated by the sound wave, $u_* + c_* = c_0$, in which, forward part of rarefaction wave traverses the head shock, the tail shock traverses the backward part, and (c) the interaction results in decaying of shocks to a Mach wave.

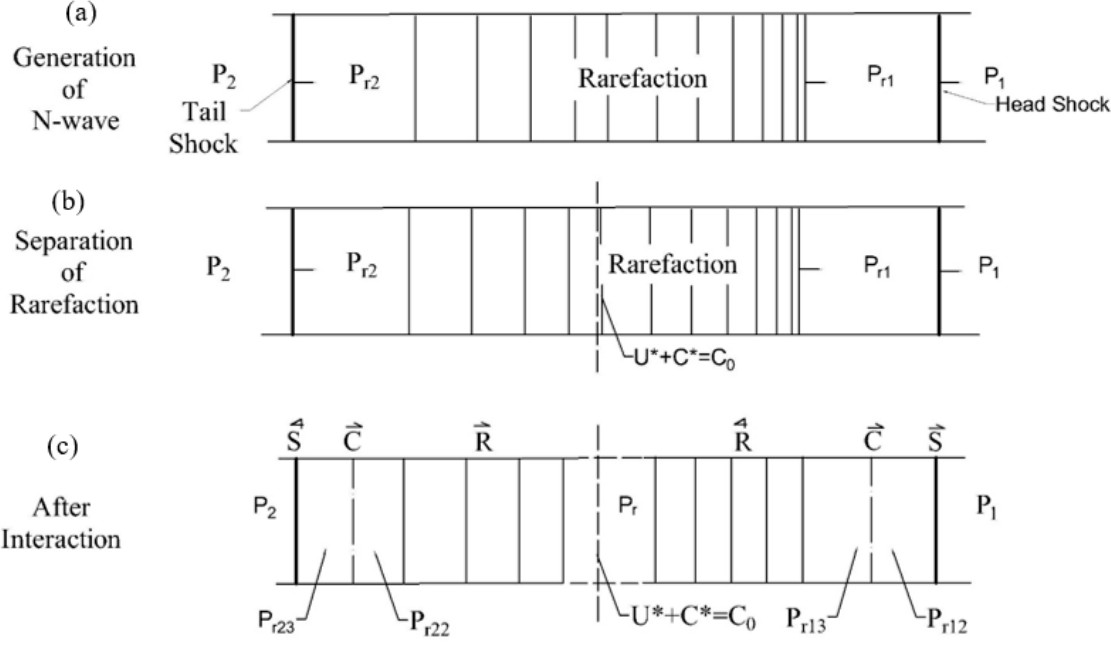

**Figure 9.** Initial and final conditions of N-wave.

## 6. Summary and Conclusions

In linear acoustics, small disturbances always propagate at a constant speed relative to the medium and do not influence the fluid properties or behavior. This situation changes dramatically if the wave intensity is increased. Familiar laws like the principle of superposition do not apply for intense waves. The local sound speed changes from $c_0$ to $c_u + u$. Thus, due to acoustic nonlinearity, propagation velocities in different regions of a waveform are different, and, shocks (discontinuities) are formed within the waveform.

In this research, the one-dimensional motion of an ideal fluid is used to study the nonlinear propagation of intense acoustic waves. With a pulse of high-intensity waves, two shocks are formed at the boundaries and a rarefaction zone of varying velocities is formed between them. The strength of the shock will depend on the power level of the source, the higher the power level stronger the discontinuity. The rarefaction can be separated into two zones separated by the sound wave, $c_u + u = c_0$. The leading edge of the rarefaction zone is traveling faster than that of the shock at the front, while the second shock at the back is travelling at a faster speed than that of the trailing edge of the rarefaction zone. The leading edge of rarefaction will overtake the shock at the front, while the tail shock overtakes the trailing edge of the rarefaction wave. The pressure jump across the shocks will continue to decay during these interactions. This results in the dissipation of the shock's energy. Since energy is conserved, it is transferred from wave energy to heat. This research helps to understand the effect of the nonlinearity of intense ultrasonic waves and can be used to model the processes involving the propagation of ultrasonic waves.

## 7. Suggestions for Future Work

In this research, an attempt is made to understand the formation of shocks within intense waves and the decaying of acoustic waves due to sound-sound interactions. However, in this study, the medium of propagation is an ideal fluid. It is proposed to extend the scope of this research to include the propagation of intense acoustic waves in non-ideal media, such as water. This will help to better understand the interactions of the wave within fluids that can be used to study the distribution of high-intensity acoustic wave in system such as sonochemical reactors and the effects of parameters such as frequency, acoustic intensity, etc. In a pulse cycle, the shock will not appear immediately adjacent to the source but will form at some distance from it, and the shock will continue to decay along the direction of propagation as the interaction with the rarefaction occurs. Determining the distance from the source at which the shock appears and until when its decays will be essential to improving the applications of intense ultrasound.

**Author Contributions:** Conceptualization, J.A.K.; formal analysis, Z.Z.; investigation, J.A.K., R.W.M., and J.N.M.; methodology, B.G.B. and J.N.M.; supervision, B.G.B. and J.N.M.; validation, Z.Z., B.G.B., and J.N.M.; writing—original draft, J.A.K.; writing—review and editing, Z.Z., R.W.M., and J.N.M. All authors have read and agreed to the published version of the manuscript.

**Funding:** This research was sponsored by the US National Science Foundation Award #1634857 entitled "Remediation of Contaminated Sediments with Ultrasound and Ozone Nano-bubbles."

**Conflicts of Interest:** The authors declare no conflict of interest.

## Appendix A

The following equation can be used to calculate the maximum displacement.

$$A = \sqrt{\frac{2I}{(\rho c)\omega^2}} = \sqrt{\frac{2 \times (8 \times 1500)/(3.14 \times 25 \times (12.7 \times 10^{-3})^2}{(1.225 \times 343) \times 12560000^2}} = 1.69 \times 10^{-5} m$$

$$\omega = 2 \times \pi \times f = 2 \times 3.14 \times 2 \times 10^6 = 12560000 \; Hz$$

$$u = 2 \times A \times f \times \pi = 1.69 \times 10^{-5} \times 12560000 \approx 212.39 \; m/s$$

Using Equations (4) and (7) to calculate the shock speed and density behind the shock.

## Appendix B

Using Hugoniot Equation (7):

$$\frac{1.4 \times \left(\frac{1}{1.225}\right) \times 101325}{1.4-1} - \frac{1.4 \times \left(\frac{1}{\rho_{r1}}\right) \times 229029}{1.4-1}$$
$$= \frac{(101325-229029)\left(\left(\frac{1}{1.225}\right)+\left(\frac{1}{\rho_{r1}}\right)\right)}{2}$$

$$\rho_{r1} = 2.16 \; kg/m^3$$

With the conservation of mass, Equation (4):

$$2.16(212.39 - U_1) = 1.225(0 - U_1)$$

$$U_1 = 490.65 \; m/s$$

## Appendix C

Equation (15) is used to calculate the pressure jump across the tail shock. As shown in Figure 4, the subscripts "r2" and "2" represent the fluid ahead of and behind the shock, respectively. Where $u_2$

is the particle velocity behind the tail shock, which is 0 $m/s$, $u_{r2}$ is $-212.39$ m/s, $P_{r1}$ is $40,055$ Pa, $\rho_{r2}$ is $0.62\,\mathrm{kg/m^3}$, and $\gamma$ for normal air equals 1.4. Substituting these values into Equation (15), the unknown $P_2$ can be calculated:

$$u_2 = u_{r2} + \frac{\sqrt{\frac{2P_{r2}}{\rho_{r2}}}\left(\frac{P_2}{P_{r2}}-1\right)}{\left[\left\{(\gamma_r+1)\frac{P_2}{P_{r2}}\right\}+(\gamma-1)\right]^{1/2}}$$

$$0 = -212.39 + \frac{\sqrt{\frac{2\times40055}{0.62}}\left(\frac{P_2}{40055}-1\right)}{\left[(1.4+1)\frac{P_2}{40055}+(1.4-1)\right]^{1/2}}$$

$$P_2 = 99,954 \ or \ 13,818 \ Pa$$

For the propagation direction, right, the particle speed $u_{r2}$ and sound speed $c_{r2}$ are in opposite direction and $-u_{r2}+c_{r2}$ is smaller than $c_2$. Therefore, $P_{r2} < P_2$, and $P_2$ has the value 99,954 Pa.

Equations (7) and (10) can be used to calculate the shock speed and density behind the shock. Using the Hugoniot expression as in Equation (10) to calculate the density behind the shock gives.

$$\frac{\frac{1.4\times\left(\frac{1}{0.62}\right)\times40055}{1.4-1} - \frac{1.4\times\left(\frac{1}{\rho_2}\right)\times99954}{1.4-1}}{\frac{(40055-99954)\left(\left(\frac{1}{0.62}\right)+\left(\frac{1}{\rho_2}\right)\right)}{2}}$$

$$\rho_2 = 1.38 \ \mathrm{kg/m^3}$$

With the conservation of mass, Equation (7) shock speed can be find:

$$1.38(0 - U_1) = 0.62(-212.39 - U_2)$$

$$U_2 = 173.27 \ \mathrm{m/s}$$

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
