# Peer review of "Nonlinear Behavior of High-Intensity Ultrasound Propagation in an Ideal Fluid"

_acoustics, doi:10.3390/acoustics2010011_

Round 1

Reviewer 1 Report

The work proposed a method to study the nonlinearity associated with intense ultrasound by using the 12 one-dimensional motion of nonlinear shock wave in an ideal fluid. The work sounds good but some critical revisions are needed

 (1) The main contribution of this paper is not clear. The equations used to compute the ultrasound waves are based on other researches like references [10], [11], etc.. The contribution of the present work should be emphasis.

(2) It is not clear why the nonlinear properties including the decaying of the N-wave in Section 3 are important in engineering practice. More explanations are needed.

(3) More details should be explained in the simulation study to show how these simulation results validate the contributions of the study

The manuscript is acceptable after a revision.

Author Response

Manuscript ID: acoustics-723181
Type of manuscript: Article
Title: Nonlinear Behavior of High-Intensity Ultrasound Propagating in an Ideal Fluid

Authors: Jitendra Kewalramani *, Zou Zhenting, Richard Marsh, Bruce Bukiet, jay Meegoda

The authors would like to acknowledge all the reviewers for their insights into the work presented in this article and the editor for expediting the reviews. The comments have been taken into consideration, and changes were made to the manuscript to reflect the comments and suggestions of the reviewers. Responses to individual comments can be reviewed below.

  • Note: All author responses are in bold, italics, and bulleted.

Reviewers' comments:

Reviewer 1

The work proposed a method to study the nonlinearity associated with intense ultrasound by using the 12 one-dimensional motion of nonlinear shock wave in an ideal fluid. The work sounds good, but some critical revisions are needed

 (1) The main contribution of this paper is not clear. The equations used to compute the ultrasound waves are based on other researches like references [10], [11], etc. The contribution of the present work should be the emphasis.

  • In this manuscript, the mathematical equation for nonlinear wave motion is used to model and understand the nonlinear phenomena associated with high-intensity acoustic waves. The manuscript is modified and included the above.

(2) It is not clear why the nonlinear properties including the decaying of the N-wave in Section 3 are important in engineering practice. More explanations are needed.

  • The introduction is modified to include a description of power ultrasonics and its many applications. The importance of modeling this behavior is specifically outlined in lines 49-51, 59-61 and 411-413. Essentially, understanding the basic mechanisms of the nonlinear effects of intense acoustic waves is critical to practical applications of power ultrasonics such as cleaning, enhancing chemical reactions, emulsification, etc.

(3) More details should be explained in the simulation study to show how these simulation results validate the contributions of the study

  • In the manuscript, a pulse ultrasonic wave is conceived as one complete cycle of piston motion in a tube. In the revised manuscript, the description of analogy is further explained in lines 104-116 and figure 3 is added.

Reviewer 2 Report

In this paper, the one-dimensional motion of an ideal fluid is used to study the nonlinear propagation of intense acoustic waves.  Using an acoustic pulse, two shocks are formed at the boundaries and a rarefaction zone of varying velocities is formed between them. Interaction between the two shock waves and the rarefaction waves are described.

The introduction doesn’t provide sufficient references to previous literature. Probably there have been papers on the topics, that have to be cited to judge the current paper originality. Nevertheless the paper is interesting but the quality of presentation has to be improved.

I add a file for some improvements of the paper clarity and some questions.

Author Response

Reviewer 2 comments:

In this paper, the one-dimensional motion of an ideal fluid is used to study the nonlinear propagation of intense acoustic waves.  Using an acoustic pulse, two shocks are formed at the boundaries and a rarefaction zone of varying velocities is formed between them. Interaction between the two shock waves and the rarefaction waves are described.

The introduction doesn’t provide sufficient references to previous literature. Probably there have been papers on the topics, that have to be cited to judge the current paper originality. Nevertheless, the paper is interesting but the quality of the presentation has to be improved.

Reply:

The author would like to acknowledge all the reviewers for their insights into the work presented in this article and the editor for expediting the reviews. The comments have been taken into consideration, and changes were made to the manuscript to reflect the comments and suggestions of the reviewers. Responses to individual comments can be reviewed below.

  • In revised manuscript the introduction is modified. The first section of manuscript is split into two sections: (1) Introduction and (2) Nonlinear wave propagation and shock formation.
  • In the revised manuscript the previous literature are cited appropriately
  • Corrections to language and style are made throughout the paper and comments in the provided document are addressed, as indicated by tracked changes in the word document.
